# Brain Structural and Functional Alterations in Multiple Sclerosis-Related Fatigue: A Systematic Review

Chiara Barbi [1], Francesca Benedetta Pizzini [2,*], Stefano Tamburin [1], Alice Martini [3], Anna Pedrinolla [1], Fabio Giuseppe Laginestra [1], Gaia Giuriato [1], Camilla Martignon [1], Federico Schena [1] and Massimo Venturelli [1,4]

1 Department of Neurosciences, Biomedicine, and Movement, University of Verona, 37100 Verona, Italy; chiara.barbi@univr.it (C.B.); stefano.tamburin@univr.it (S.T.); anna.pedrinolla@univr.it (A.P.); fabiogiuseppe.laginestra@univr.it (F.G.L.); gaia.giuriato@univr.it (G.G.); camilla.martignon@univr.it (C.M.); federico.schena@univr.it (F.S.); massimo.venturelli@univr.it (M.V.)

2 Radiology, Department of Diagnostic and Public Health, University of Verona & Department of Diagnostics and Pathology, University Hospital, 37129 Verona, Italy

3 School of Psychology, Keele University, Newcastle ST5 5BG, UK; a.martini@keele.ac.uk

4 Department of Internal Medicine, University of Utah, Salt Lake City, UT 84112, USA

* Correspondence: francesca.pizzini@aovr.veneto.it

**Abstract:** Fatigue is one of the most disabling symptoms of multiple sclerosis (MS); it influences patients' quality of life. The etiology of fatigue is complex, and its pathogenesis is still unclear and debated. The objective of this review was to describe potential brain structural and functional dysfunctions underlying fatigue symptoms in patients with MS. To reach this purpose, a systematic review was conducted of published studies comparing functional brain activation and structural brain in MS patients with and without fatigue. Electronic databases were searched until 24 February 2021. The structural and functional outcomes were extracted from eligible studies and tabulated. Fifty studies were included: 32 reported structural brain differences between patients with and without fatigue; 14 studies described functional alterations in patients with fatigue compared to patients without it; and four studies showed structural and functional brain alterations in patients. The results revealed structural and functional abnormalities that could correlate to the symptom of fatigue in patients with MS. Several studies reported the differences between patients with fatigue and patients without fatigue in terms of conventional magnetic resonance imaging (MRI) outcomes and brain atrophy, specifically in the thalamus. Functional studies showed abnormal activation in the thalamus and in some regions of the sensorimotor network in patients with fatigue compared to patients without it. Patients with fatigue present more structural and functional alterations compared to patients without fatigue. Specifically, abnormal activation and atrophy of the thalamus and some regions of the sensorimotor network seem linked to fatigue.

**Keywords:** multiple sclerosis; fatigue; neuroimaging; MRI

## 1. Introduction

Multiple sclerosis (MS) is an inflammatory demyelinating autoimmune disease of the central nervous system (CNS) [1]. Atlas of MS 2013 has estimated an increase in the number of people affected by MS in the world from 2.1 million in 2008 to 2.3 million in 2013 [2]. Pathological features of MS include autoreactive immune cells attacking axons and myelin of CNS neurons. Specifically, this characteristic causes lesions in the brain and the spinal cord which all contribute to sensory, motor, and cognitive symptoms and autonomic dysfunctions [3]. MS's pathogenesis is still debated. It seems that a complex interplay between environmental and genetic factors plays a key role in the nature of MS. Moreover, chronic cerebrospinal venous insufficiency was identified as a possible factor underlying pathogenesis of MS [4]. The age at disease onset is usually between 20 and 40 years [5].

Late-onset (50 years or more) is not rare and presents similar neurological presentation to early-onset. On the other hand, the progression to disability is more rapid [6]. The early stage of MS is characterized by relapses followed by a full recovery. This stage is called the relapsing-remitting phase (RRMS) [7]. The gradual increase of disability independent of relapses over time characterizes the progression of disease and the other clinical form of MS called progressive MS—first of all, with primary progressive form, followed by the secondary progressive MS [8,9].

One of the most disabling symptoms for patients with MS is chronic fatigue [10]. Fatigue is defined as a subjective sensation of weariness, increasing sense of effort, mismatch between the effort spent and actual performance, or exhaustion [11]. There is also an objective definition of fatigue: the concept of fatigability. It is important to note that there is an important difference between the perception of fatigue and fatigability. Although fatigue is defined as subject sensation, fatigability is the magnitude of change in a performance criterion over a given time of movement task. Indeed, the perceptions of fatigue and fatigability are not only distinct but also potentially independent [11]. This symptomatology is reported in around 70–80% of patients with MS. Moreover, fatigue is the most disabling symptom for 55% of patients and is associated with lower quality of life [12]. The nature of fatigue could either be primary or secondary to other variables [13] (Figure 1). In the first case, fatigue is a direct consequence of disease and its processes. It seems that the peripheral and central immunological and inflammatory process might play a central role in the exacerbation of fatigue, specifically in patients with MS [14]. Indeed, levels of cytokines play a key role in pathogenesis of MS. It is well known that pro-inflammatory cytokines operate directly on the brain to induce sickness behavior, reduced motivation, increased pain sensitivity, evident fatigability, and depressed mood [14,15]. They act affecting the monoaminergic neurotransmission and damaging the mesocorticolimbic pathways (crucial for valence and reward processing) [16]. Moreover, the levels of interleukin 6 are related with relapse and remission phases, which are strongly associated with fatigue [17]. It is important to know that immune activation is correlated to changes in neuroendocrine function, causing fatigue in patients with MS. Other relevant co-factors are endocrine and neurotransmitter dysregulation. They play a key role in exacerbation of fatigue, but is not clear whether the endocrine element is a primary or secondary cause of fatigue [18]. The persistent endocrine and autonomic disturbances are likely due to gray matter (GM) lesion in the hypothalamus or brainstem nuclei that could disturb the hypothalamus-pituitary-adrenal axis and descending neural control of the autonomic nervous system [19]. Indeed, the autonomic nervous system dysfunction in patients with MS appears involved in the exacerbation of the symptom of fatigue [19–23]. Dinoto et al. [24] reported a strong correlation between fatigue and autonomic nervous system dysfunction in patients with MS. Specifically, they found that patients with fatigue had a significantly higher dysautonomia compared to patients without fatigue. Indeed, it seems that the autonomic nervous system is regulated by the same brain areas involved in the perception of fatigue. Further, the vagus nerve (the connection between interoceptive areas and autonomic involvement) is affected by pro-inflammatory cytokines, and its overactivation connects the symptom of fatigue and autonomic dysregulation [20].

On the other hand, secondary fatigue may result from other symptoms, such as level of disability, sleep problems, depression, reduced activity, or from medication use [13]. Indeed, pain medications, antispasticity agents, sedatives, anticonvulsants, and antihistamines have as a side effect of fatigue. Moreover, physical pain, sensory disturbances—such as dysesthesia and neuralgia—and painful muscle spasms induce physical deconditioning, sleepiness, and depression, which have a strong relation with the symptom of fatigue [18]. Indeed, more than half of patients with MS report symptoms of fatigue together with symptoms of depression and pain [16]. The coexistence of these three symptoms suggests a common etiology. Specifically, they are an important sign of anhedonia (decreased ability to attempt for and to experience pleasure) [25,26], which has been imputed to deficits in reward processing [27] and is a central component of emotional responses, behavior, and

learning [16]. The shared etiology was demonstrated by several studies [28–32]. Seixas et al. [28] reported functional and structural alteration in the brain structures implicated in the reward circuitry in patients with MS that reported chronic pain, specifically in the caudate nucleus, the nucleus accumbens, and the mesial temporal lobe. The ventral striatum, including the nucleus accumbens and the caudate nucleus, is associated with the limbic structures and the prefrontal cortex and is implicated in motivational and emotional aspects of behavior, including reward. Moreover, GM atrophy in the basal ganglia, primarily the striatum and the limbic system, was shown in patients with MS who reported fatigue and depression [16].

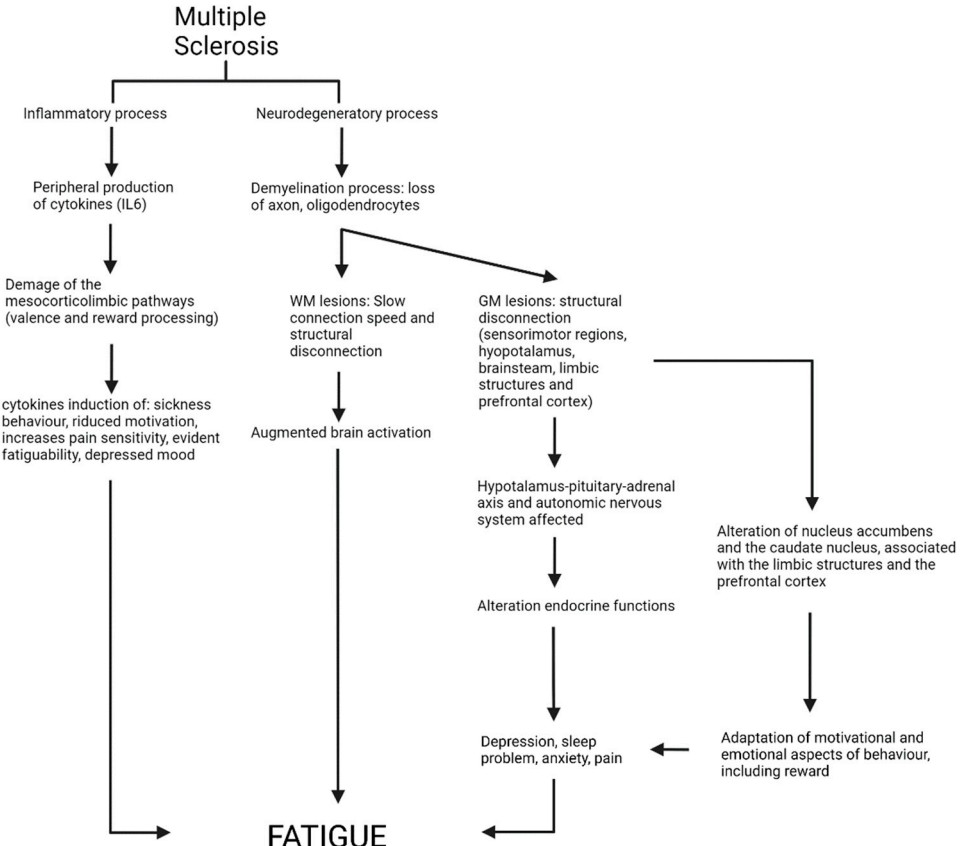

**Figure 1.** Etiology of fatigue in MS.

Recently, some studies have provided potential mechanisms underlying the subjective experience of fatigue [33–36], such as metacognitive mechanisms [14]. An interesting one focuses only on the sensorimotor system [34]. Since patients with MS present diminished sensory attenuation, the movement execution brings the brain to conclude that the execution demands more effort than predicted [14]. This theory supposes that fatigue is a straight consequence of unexpectedly high observed effort [14]. Unfortunately, the pathogenesis of MS-induced fatigue is complex and not fully understood.

Despite previous studies investigating the association between several factors, such as: depression, cognitive impairments, medications, proinflammatory cytokines, cerebral structural defects, altered patterns of cerebral activation, endocrine abnormalities, axonal injuries, and the presence of fatigue in patients with MS, the nature of this phenomenon is still not completely clear [37–44]. Fatigue is usually evaluated with a large variation of self-reported questionnaires in the clinical setting [45–48]. Although this approach has been extensively utilized, some limitations need to be accounted for, such as the lack of specificity about the nature of these symptoms. Moreover, in clinical practice the use of a reliable and standardized fatigue scale is essential to plan and supervise an adequate personalized treatment strategy [49]. However, the large scale heterogeneity and a missed

consensus on management of fatigue make the control of this symptom in patients with MS challenging [49].

The advanced technology applied to neuroimaging, such as magnetic resonance imaging (MRI), functional magnetic resonance imaging (fMRI), and positron emission tomography (PET), could provide important results in order to better understand the nature of fatigue. Indeed, neuroimaging techniques may highlight associations between structural and functional cerebral dysfunctions and symptoms of fatigue in patients with MS. (Figures 2 and 3) The structural information provided by MRI is the gold standard in the diagnosis of MS. Recently, researchers utilized a combination of structural and functional imaging (e.g., fMRI, PET) in order to better understand the development of MS. Several papers support the idea that the structural white matter (WM) and GM lesions disseminated in space and in time have a potential link with the symptom of fatigue [14]. On the other hand, considering comparative studies between patients with fatigue (F) and patients without fatigue (NF), they suggest that there is a lack of difference in terms of structural parameters between the two patient groups [50–59]. Examing studies that use functional methodologies, it seems that there are functional brain differences between F and NF patients [60–69]. Namely, patients with fatigue reported an increase of distributed brain activity during the performance of tasks [14]. Considering only the sub-domain of cognitive fatigue, structural differences in the subcortical region were identified in patients with cognitive fatigue (CF) [70–72]. Since a general consensus of the etiopathogenesis of fatigue in patients with MS is missing, this systematic review aims to understand whether structural and functional brain damage revealed by neuroimaging correlates with fatigue in patients with MS.

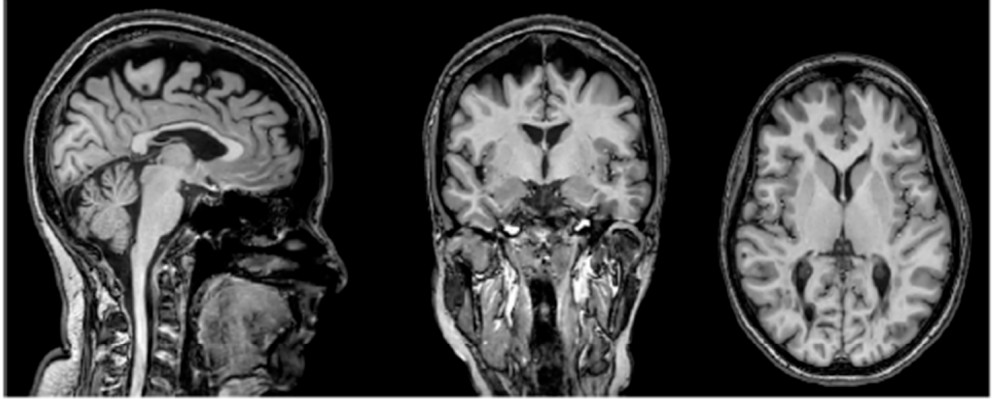

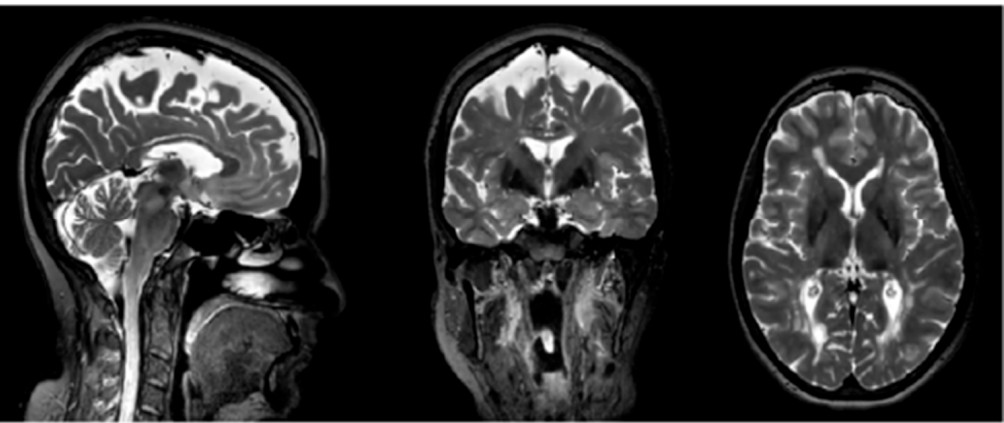

**Figure 2.** *Cont.*

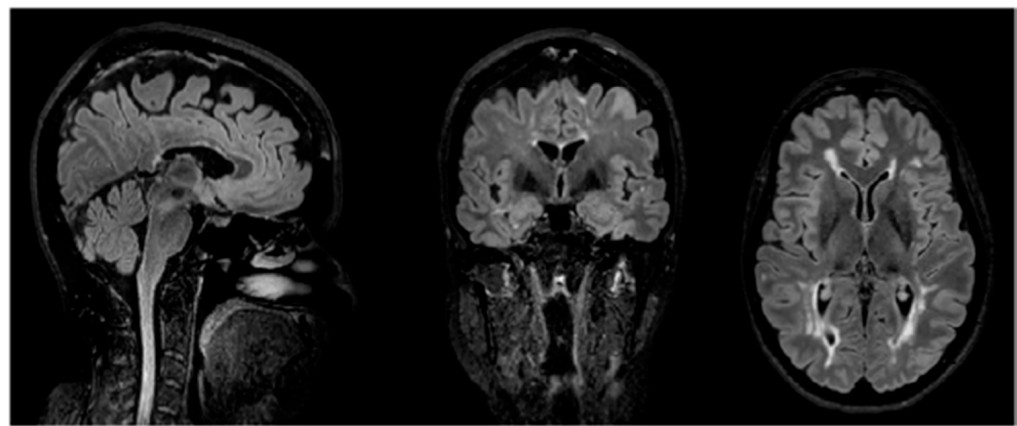

**Figure 2.** Multiplanar reconstruction of T1 (**top**), T2 (**middle**), and FLAIR (**bottom**) volumetric acquisitions showing multiple demyelinated lesions, confluent posteriorly.

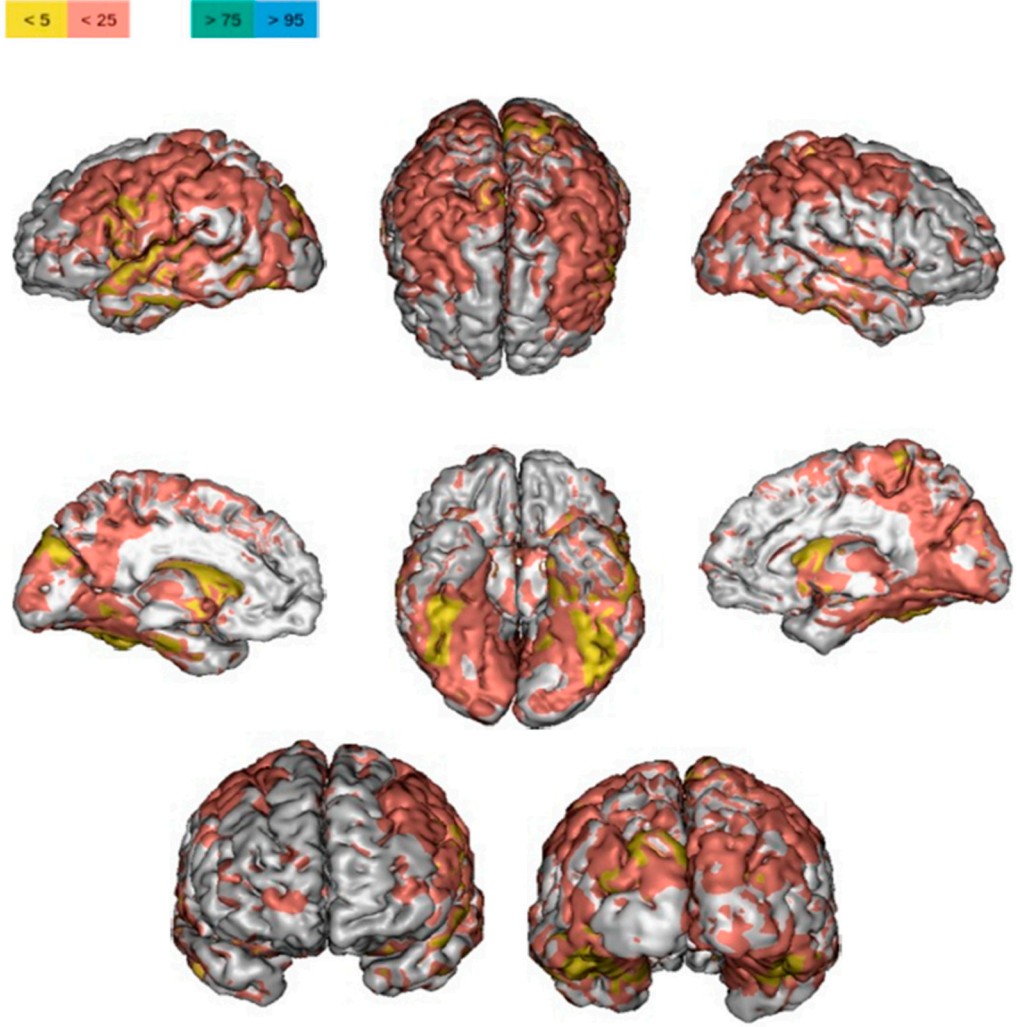

**Figure 3.** Example of volume rendering of brain volume analysis in MS (volume, %ICV, percentile). The percentile ranges to which each brain area belongs are highlighted in four colors. In this MS patient, there are broad areas of atrophy (<5th and 25th percentiles, respectively, in yellow and pink), particularly affecting fronto-parietal and temporal lobes. Powered by QyScore®.

## 2. Materials and Methods

Conforming to PRISMA (Preferred Reporting Items for Systematic reviews and Meta-Analyses) guidelines [73], we executed Medline searches to determine all neuroimaging studies of fatigue in MS from 1980 until February 2021. The systematic review has been registered with the code CRD42022333610. After duplicate exclusion, 1437 studies were included in the title and abstract screening. After limiting the results by criteria described below, 50 studies were considered eligible to enter the systematic review (Figure 4).

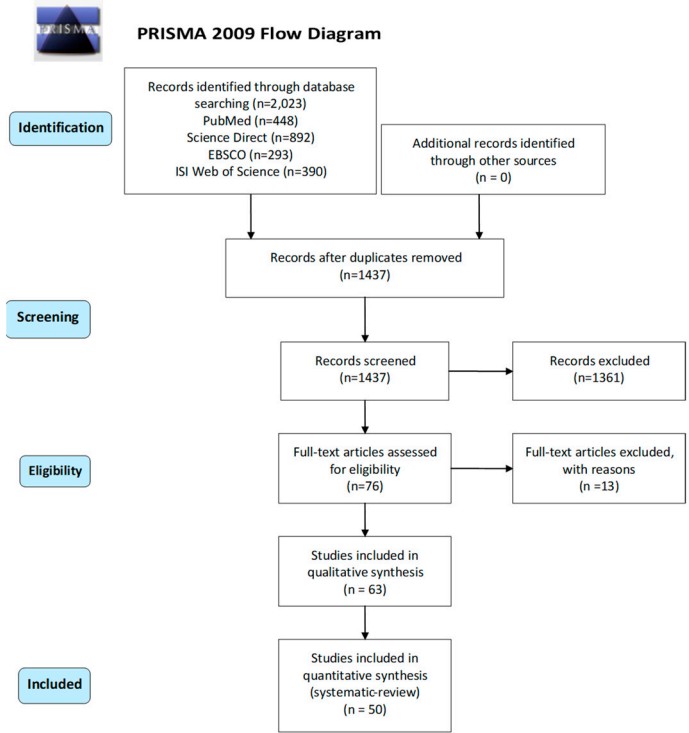

**Figure 4.** PRISMA flow diagram. From: Moher D, Liberati A, Tetzlaff J, Altman DG, The PRISMA Group (2009). Preferred Reporting Items for Systematic Reviews and Meta-Analyses: The PRISMA Statement. PLoS Med 6(6): e1000097. doi:10.1371/journal.pmed1000097. For more information, visit www.prisma-statement.org [73].

Eligibility criteria were:

**Population:** Only studies with comparisons between F and NF patients (regardless of MS sub-types—such as relapsing-remitting (RRMS), primary progressive (PPMS), secondary progressive (SPMS)—and time since disease diagnosis) were included. The studies with comparison only between MS and HC were excluded. Fatigue in patients was assessed using a validated clinical questionnaire and published cutoff scores for fatigue in MS. All sub-domains of fatigue were considered.

**Participants:** Female and male adults were included; pediatric patients were excluded due to physiological differences. Animal studies were excluded.

**Intervention:** All studies using functional and structural imaging that aimed to study fatigue symptoms were included. Studies that reported only association or correlations between fatigue score and neuroimaging results were excluded.

**Other criteria:** Language: Studies written in English were included. Conference proceedings and unpublished studies were excluded.

**Search strategy:** Electronic databases were autonomously searched by the researchers (A.M., C.B.) from 1980 until September 2017. Another update of research was made by CB from 1980 to 24 February 2021. The following electronic databases were selected: PubMed, Science Direct, EBSCO, ISI Web of Science.

Pre-defined search strings with Boolean operators included: *Multiple sclerosis AND fatigue AND voxel-based OR morphometry OR VBM OR MRI OR structural magnetic resonance imaging OR diffusion spectrum imaging OR diffusion MRI OR DTI OR DSI OR diffusion magnetic resonance imaging OR fMRI OR PET OR SPECT OR functional magnetic resonance imaging OR functional MRI OR neuroimaging.*

In the first search, title and abstract screening was performed, independently, by two authors (C.B., A.M.) using Rayyan QCRI program [74]. In the second search, title and abstract screening was performed by CB using Rayyan QCRI program [74]. In the first search, two authors (C.B., A.M.) independently evaluated papers selected for full-text examination. In the second search, evaluated papers were selected for full-text examination. In the first search, disagreements were resolved after discussion with a third researcher (S.T.). In the first search, the following data were extracted, independently, by A.M. and C.B.: demographical and clinical information: sex, age, type of MS, MS duration, expanded disability status scale (EDSS), depression, and cognitive evaluation (reported in Table S1); methods: imaging technique matched and unmatched variables and results. In the second search, data extraction was performed, independently, by C.B.

Any differences in terms of structural or functional measures were eligible for inclusion. Results could be reported as global brain differences between patients, or specific brain regions or specific networks could be compared between patients. Longitudinal studies were included; no restrictions were placed on the number of points at which the outcomes were measured. Where multiple comparisons were reported, including comparison with healthy control, only outcomes regarding differences between patients were considered.

The variables collected for which data were sought were:

- The report: author, year, journal;
- The study: participants' characteristics, definition and criteria for fatigue;
- The participants: sex, age, education, EDSS, MS type, diagnosis criteria, MS duration, medications, other symptoms;
- The research design: scan design;
- The intervention: imaging technique, scanner type, smoothing, software analysis.

## 3. Results

### 3.1. Search Results

Fifty studies were included in this systematic review. Nine of them analyzed structural brain damage in WM by comparing F and NF [39,53–55,57,58,75–78], and four papers assessed WM alterations between CF and CNF patients [70–72,79]. Twelve studies examined structural brain damage in GM comparing F and NF patients [39,40,50–52,56,59,72,80–88]. Nine studies were included in both GM and WM alterations sections [39,52,53,56,61,76,77,86,87]. One paper reported structural alterations in WM and GM comparing patients with and without cognitive and global fatigue [89]. Thirteen papers reported functional alteration [60,62–69,90–95], and four included both structural and functional brain damage in F and NF [61,82,91,96]. One paper studied the differences in terms of functional alteration in CF and CNF patients [97].

### 3.1.1. Structural Neuroimaging Findings Correlated to Fatigue

**Conventional MRI and atrophy**: Three studies assessed WM lesion load (LL) using a semi-automated thresholding technique in 3D-Slicer version 3.4, and two studies examined WM volumes obtained from 3D T1 images using the unified segmentation approach of statistical parametric mapping (SPM) 8. Two studies reported T2 hyperintense and T1 hypointense lesion volume (LV) measured on DE TSE and 3D T1-weighted scans. Moreover, they assessed WM volumes using SIENAx [89].

Using voxel-based morphometry (VBM), one study reported a higher WM atrophy in F compared to NF patients [61]. On the other hand, one study did not find differences between the two patient groups [56]. One study reported no differences in terms of WM LL tracts between F and NF patients [57].

Four papers reported a higher value of LL in F patients [52,82,86,87]. On the other hand, three studies did not find any differences in terms of lesion distribution and LV between two groups of patients [40,51,52]. The LV resulted higher in F compared to NF patients in two studies [55,82]. One study did not find any differences in terms of T2, T1 LV, or in WM volume between F and NF patients and between CF and CNF [89].

Twenty-two cross sectional studies reported results from cortical and subcortical volume. Only one study described the differences between CF and CNF patients (further details: Tables 1 and 2).

Eight studies described reduction of global cortical volume [40,53,61,80,81,84,86]. One paper assessed the reduction of GM density [82]. One study reported no differences between two patient groups in terms of volume reduction of GM [56]. Three studies reported a reduction of volume in F compared to NF patients [50,76,81].

Three studies reported a reduction of cortical thickness (Cth) in F patients [50,72,85]. Two papers did not find any differences between F and NF in terms of global Cth [51] and Cth in rolandic regions [59].

Six studies reported reduction of volume in F and NF patients compared to HC [40,50,61,76,86,96]. The Cth resulted significantly lower in F patients compared to HC [72].

**DWI**. Ten papers used diffusion-weighted images (DWIs) in order to analyze subcortical WM tracts. Using diffusion tensor imaging (DTI), three cross-sectional studies and one longitudinal study reported WM differences between CF and CNF patients. Five studies reported a lower FA in F than NF patients. [55,75–78].

Considering the sub-domain of cognitive fatigue only, one study reported a lower value of FA in left amygdala in CNF than CF patients [71].

Two studies assessed RD (radial diffusivity) value, and only one reported a higher value of RD in F than NF [77]. RD values resulted lower in CF than CNF patients in two papers [70,98].

Fours studies reported MD (mean diffusivity); only one found a higher value in F than NF [75]. Four papers did not find any differences between two groups of patients in terms of MD [55,64].

Axial diffusivity (AD) resulted lower in CF than CNF patients [70,98].

The longitudinal study reported higher values of AD and RD in F compared to NF patients after 17 months [79].

In terms of magnetization transfer ratio (MTR), two studies reported similar results between F and NF patients [53,54].

WM atrophy was higher in both groups of patients compared to HC [61] (further details in Tables 3 and 4).

### 3.1.2. Functional Neuroimaging Findings Correlated to Fatigue

Using resting state fMRI (rs-fMRI), five cross-sectional studies reported measures of functional connectivity (FC) [60,61,63,69,96]; eight others indicated differences in terms of activation during task-based fMRI between F and NF patients [62,64–66,68,90–92]. Two studies assessed brain metabolism using resting state positron emission tomography (PET) [67,82]; three other reported brain metabolites N-acetylaspartate (NAA) and creatine (Cr) using proton MR spectroscopic imaging (MRSI) in F and NF patients [93–95].

Only one study reported differences in terms of brain connectivity between CF and CNF patients using task-based fMRI [97] (further details in Tables 5 and 6).

**Table 1.** Key details of the structural studies on WM in MS patients with fatigue including imaging techniques, subjects, and outcome.

| Reference | Imaging Technique | Subjects | Fatigue Scale | Matched Variables | Unmatched Variables | Neuroimaging Findings Correlated to Fatigue | Findings: F, NF vs. HC | Findings: F vs. NF |
|---|---|---|---|---|---|---|---|---|
| *Cross-sectional* | | | | | | | | |
| [53] | DTI | F:17 NF:17 | FSS | Disease duration, age, sex, immunomodulatory treatment, DSC score. | EDSS, central motor activation. | DTI FA, DTI ADC, MTR | | F = NF |
| [54] | DTI | F:30 NF:30 | FSS | Age *, sex *, disease duration, education, EDSS, PASAT, T2-LV, NBV, NGMV, pharmacological treatment. | | FA, MD, RD, and AD | | F = NF |
| | | | | | | **FA** Frontal and occipital U-fibers, R external capsule, L uncinate fasciculus, forceps minor, L superior longitudinal fasciculus, bilateral cingulum, and pons ($p \leq 0.05$) | F↓ | |
| | | | | | | **MD, RD** Frontal and occipital U-fibers, right external capsule, L uncinate fasciculus, forceps minor, L superior longitudinal fasciculus, bilateral cingulum, and pons ($p \leq 0.05$) | F↑ | |
| | | | | | | **AD** L internal capsule, bilateral external capsule, bilateral corona radiata, L superior longitudinal fasciculus, bilateral anterior thalamic radiation, R inferior fronto-occipital fasciculus, and forceps minor ($p \leq 0.05$) | NF↑ | |
| [77] | DTI, volume of subcortical nuclei, and brainstem structures. | F:15 Moderately F:14 NF:14 | FSS | Age, disease duration, pharmacological treatment, EDSS, T2 LV | | Volume of thalamus ($p = 0.001$), pallidum ($p = 0.013$), and superior cerebellar peduncle ($p = 0.002$). | | F↓ |
| | | | | | | RD in R temporal cortex ($p = 0.016$, corrected $p = 0.026$) FA in R temporal cortex ($p = 0.004$, corrected $p = 0.005$) | | F↑ F↓ |
| [54] | MT and DT MRI | F:14 NF:14 | FSS | Age, disease duration, EDSS | | MTR, FA, and MD | | F = NF |
| [39] | MRI | F:15 NF:15 | FSS | Age, sex, disease duration, EDSS pyramidal score, MADRS | | Median MRI total lesion burden the parietal lobe ($p < 0.05$), internal capsule ($p < 0.05$), and periventricular areas ($p < 0.05$). | | F↑ |
| [82] | VBM | F:11 NF:6 | EMIF-SEP | Age, sex, EDSS, disease duration, MADRS, Mattis score, lesion volume | | LV: juxtacortical and/or overlapping cortico-subcortical lesions located in frontal and temporal areas ($p < 0.05$). | | F↑ |
| [55] | DT MRI | F:81 NF:66 | FSS * | Sex, age, disease duration, PASAT, pharmacological treatment, T2 LV, T1 LV, NBV, NGWV, NWMV | EDSS, MADRS * | MD | | F = NF |
| | | | | | | FA of the Fm ($p = 0.02$), R ATR ($p = 0.03$) | | F↓ |

**Table 1.** *Cont.*

| Reference | Imaging Technique | Subjects | Fatigue Scale | Matched Variables | Unmatched Variables | Neuroimaging Findings Correlated to Fatigue | Findings: F, NF vs. HC | Findings: F vs. NF |
|---|---|---|---|---|---|---|---|---|
| *Cross-sectional* | | | | | | | | |
| [56] | VBM | F:64 NF:59 | FSS * | Sex, age, disease duration, pharmacological treatment, PASAT, T2 LV, T1 LV, NBV | EDSS, MADRS * | WM atrophy: Ant Thal Rad, Post Thal Rad, Sup Cor Rad, Post Cor Rad, cingulum, corpus callosum, SLF, ILF, IFOF, fornix, Fm, CST, cerebral peduncle, medial lemniscus, SCP, MCP, ICP regional | | F = NF |
| [61] | VBM | F:32 NF:28 | FSS * | Sex, age, disease duration, T1 LV, ICV | EDSS, CDMI | WM atrophy: L frontal areas that included the L medial frontal gyrus of the SMA, L superior frontal gyrus; L precuneus, bilateral brainstem; L and WM of the L cerebellum ($p < 0.001$) WM atrophy: bilateral frontal lobe, R middle cingulate gyrus, bilateral posterior cingulate gyrus, bilateral temporal and occipital lobes, around L thalamus and bilateral corpus callosum ($p < 0.001$) WM atrophy: frontal region (motor areas and insula), temporal, occipital, and parietal lobes. Bilateral thalamus, bilateral corpus callosum, cingulate gyrus (anterior, middle and posterior parts), bilateral brainstem and cerebellum ($p < 0.001$). | F↑<br><br>NF↑<br><br>F↑ | |
| [89] | MRI | F: 174 NF: 192 | MFIS | Sex, education, PASAT, disease duration, | Age, MADRS, EDSS | T2 LV, T1 LV, NWMV | | F = NF |
| [52] | MRI | F:16 NF:17 | FSS | Age, disease duration, EDSS, 17-HDRS | | Frontal lobe T2-LL ($p = 0.017$) | | F↑ |
| [57] | MRI | F:27 NF:21 | MFIS | Age *, sex, disease duration, EDSS | Cognitive fatigue, physical fatigue, psychosocial fatigue, tSTAI, BDI * | T2LL corpus callosum, fornix internal capsule, corona radiata, posterior thalamic radiation, sagittal stratum, external capsule, cingulum, fasciculus | | F = NF |
| | | | | | | WMLL tracts: posterior limb of the internal capsule, retrolenticular part of the internal capsule, sagittal stratum, superior longitudinal fasciculus, and uncinate fasciculus | | F = NF |
| [96] | DT MRI | F:26 Reversible F:25 NF:42 | MFIS | Age, sex, disease duration, disease category, EDSS | CES-D, T2LV* | FA bilateral fronto-orbital and subgenual regions, R superior temporal and temporal polar regions and R temporal WM, R insular and periinsular area (including the external and extreme capsules and claustrum), bilateral anterior limb of internal capsule, bilateral precommisural striatum, R amygdala and hippocampal/parahippocampal region, and R crus cerebri (F vs. NF: $p < 0.001$; F vs. reversible: $p < 0.001$. Corrected $p$ with: age + sex + DD + EDSS + LL $p = 0.954$; corrected $p$ with age + sex + DD + EDSS + LL + CES-D $p = 0.290$) | | F ↓ Reversible F = NF |
| [58] | DWIs | F:26 Reversible F:25 NF:42 | MFIS | Age, sex, disease duration, disease phenotype, EDSS, CES-D | NR | FA, AD, MD, RD of superolateral medial forebrain bundle. | | F = NF |

**Table 1.** *Cont.*

| Reference | Imaging Technique | Subjects | Fatigue Scale | Matched Variables | Unmatched Variables | Neuroimaging Findings Correlated to Fatigue | Findings: F, NF vs. HC | Findings: F vs. NF |
|---|---|---|---|---|---|---|---|---|
| *Cross-sectional* | | | | | | | | |
| [91] | DT MR | F:20 NF:15 | FSS | Sex, age, EDSS, disease duration | NR | Cord average FA ($p < 0.0001$), | F↓ | |
| | | | | | | Cord average MD ($p = 0.001$), brain NAWM average FA ($p = 0.03$), brain NAWM average MD ($p = 0.001$), brain GM average MD ($p = 0.01$) | F↑ | |
| | | | | | | Cord average FA ($p < 0.0001$) | NF↓ | |
| | | | | | | Cord average MD ($p = 0.0009$), brain NAWM average FA ($p < 0.0001$), brain NAWM average MD ($p = 0.004$), and brain GM average MD ($p = 0.0001$). | NF↑ | |
| | | | | | | Brain NAWM average FA ($p = 0.001$) | | NF↓ |
| [76] | DT MR | F:31 NF:32 | FSS | Sex, age, disease duration, EDSS, disease clinical phenotype, pharmacological treatment, MADRS, T2 LV, T1 LV. | NR | FA Fm, L inferior fronto-occipital fasciculus, R anterior thalamic radiation ($p < 0.001$, uncorrected) | | F↓ |
| | | | | | | Occurrence of lesion in the R ATR ($p < 0.001$, uncorrected). | | F↑ |
| [86] | MRI, VBM | F:43 NF:17 | MFIS | NR | T2 LL, T1 LL. | T2 LL volume ($p < 0.001$), T1 LV ($p < 0.001$) | | F↑ |
| [87] | MRI | F:197 NF:25 | FSS | Age at onset, number of relapses, WM-f. | Age, disease duration, education, AWM-f, GM-f, T2 lesion, T1 lesion. | AWM-f ($p = 0.001$), T1-LL ($p = 0.002$), T2-LL ($p < 0.001$). | | F↑ |
| [75] | DTI | F:38 NF:41 | FSMC | Age, disease duration, EDSS, education, pharmacological treatment | NR | FA for the thalamus and basal ganglia including the caudate nucleus, globus pallidus, and putamen ($p = 0.017$) | | F↓ |
| | | | | | | MD for the thalamus ($p = 0.010$) and basal ganglia including the caudate nucleus, globus pallidus, and putamen ($p = 0.030$) | | F↑ |
| | | | | | | FA thalamus ($p < 0.001$) | F↓ | |
| | | | | | | MD thalamus ($p < 0.001$) | F↑ | |
| | | | | | | FA basal ganglia | F ($p = 0.005$) and NF ($p = 0.035$) ↓ | |
| | | | | | | FA frontal cortex | F ($p < 0.001$) and NF ($p = 0.007$) | |
| | | | | | | MD basal ganglia and frontal cortex ($p < 0.001$) | F↑ | |

\* covariate. **Legend.** AD: axial diffusivity; ADC: apparent diffusion coefficient; ATR: anterior thalamic radiation; AWM-f: abnormal white matter fraction; BDI: Beck depression inventory; CDMI: Chicago multiscale depression inventory; CES-D: Center for Epidemiologic Studies depression scale; CST: cortical spinal tract; DT: diffusion tensor; DTI: diffusion tensor imaging; DWIs: diffusion weight images; DSC: digit symbol doding; EDSS: expanded disability status scale; EMIF-SEP: validated French version of the fatigue impact scale (FIS); F: patients with fatigue; FA: fractional anisotropy; Fm: forceps major; FSMC: fatigue scale for motor and cognitive function; FSS: fatigue severity scale; GM: gray matter; 17-HDRS: 17-item Hamilton depression rating scale; ICV: intracranial volume; ICP: inferior cerebellar peduncle; IFOF: inferior fronto-occipital fasciculus; ILF: inferior longitudinal fasciculus; L: left; LL: lesion load; LV: lesion volume; MD: mean diffusivity; MADRS: Montgomery Asberg depression rating scale; MFIS: modified fatigue impact scale; tSTAI: trait part of the Spielberger state trait anxiety inventory; T1: magnetization prepared rapid acquisition gradient echo; MCP: middle cerebellar peduncle; MRI: magnetic resonance imaging; MT: magnetization transfer; MTR: magnetization transfer ratio; NBV: normal brain volume; NF: patients without fatigue; NGMV: normal gray matter volume; NAWM: normal appearing white matter; NWMV: normal white matter volume; PASAT: paced auditory serial addition test; R: right; RD: radial diffusivity; SCP: superior cerebellar peduncle; SLF: superior longitudinal fasciculus; SMA: supplementary motor area; T2LV: T2 lesion volume; VBM: voxel-based morphometry; WM: white matter; WMLL: white matter lesion load.

**Table 2.** Key details of the structural studies on WM in MS patients with cognitive fatigue, including imaging technique, patient characteristics, depression/cognitive variables, and outcome.

| Reference | Imaging Technique | Subjects | Fatigue Scale | Matched Variables | Unmatched Variables | Neuroimaging Findings Correlated to Fatigue | Findings: F, NF vs. HC | Findings: F vs. NF |
|---|---|---|---|---|---|---|---|---|
| *Cross-sectional* | | | | | | | | |
| [70] | DTI | CF:37 CNF:12 | FSS | Age, education, disease duration, EDSS, TWT, 9-HP, PASAT. | **NR** | AD ($p = 0.025$) and RD ($p = 0.033$) between posterior hypothalamus and mesencephalon | | CF↓ |
| | | | | | | AD and RD fibers of the CC ($p < 0.001$) Fibers of the CC | CF and CNF↑ | CF = CNF |
| [72] | DTI | CF:20 CNF:14 | FSMC * | Age, disease duration, MSFC, BDI, LL, BPF | EDSS * (BDI > 13 *) | AD ($p = 0.016$) and RD ($p = 0.042$) R posterior hypothalamus and the locus coeruleus. AD ($p = 0.043$) and RD ($p = 0.062$) fibers between the posterior hypothalamus and the locus coeruleus in the R hemisphere | CNF↑ | CNF↑ |
| | | | | | | AD and RD CC fibers, brainstem | | CNF = CF |
| [71] | DT MRI | CF:67 CNF:28 | FSMC | Sex, disease duration, EDSS, BPF * | Age *, BDI * | FA: L amygdala | | CNF↓ |
| | | | | | | FA posterior CC, anterior CC, L stria terminalis, R stria terminalis | CF↓ | |
| | | | | | | FA posterior CC, anterior CC, L stria terminalis, L amygdala | CNF↓ | |
| | | | | | | FA: R amygdala, R stria terminalis, L stria terminalis, anterior and posterior CC | | CF = CNF |
| | | | | | | FA anterior corpus callosum ($p < 0.001$), posterior corpus callosum ($p < 0.001$) | CF and CNF↓ | |
| [89] | MRI | CF:115 CNF:251 | MFIS | PASAT, disease duration, EDSS | Sex, age, education, MADRS | T2 LV, T1 LV, normalized WM volume | CF = CNF | |
| *Longitudinal* | | | | | | | | |
| [79] | DTI | CF:28 CNF:14 | FSMC | Sex, clinical phenotype, FSMC | Pharmacological treatment, age *, education, relapse during the evaluation period | Total brain volume (GM and WM) after 17 months ($p < 0.05$) | | F↓ |
| | | | | | | AD and RD in the CC after 17 months ($p < 0.05$) Lateral ventricle volume after 17 months ($p < 0.05$) | | F↑ F↑ |

* covariate. **Legend**; AD: axial diffusivity; BDI: Beck depression inventory; BPF: brain parenchymal fraction; CC: corpus callosum; CF: patients with cognitive fatigue; CNF: patients without cognitive fatigue; DT MRI: diffusion tensor magnetic resonance imaging; DTI: diffusion tensor imaging; EDSS: expanded disability status scale; FA: fractional anisotropy; FSMC: fatigue scale for motor and cognitive function; FSS: fatigue severity scale; GM: gray matter; 9-HPT: 9-hole peg test; LL: lesion load; LV: lesion volume; MD: mean diffusivity; MFIS: modified fatigue impact scale; NR: not reported; PASAT: paced auditory serial addition test; R: right; RD: radial diffusivity; TWT: timed walk test.

**Table 3.** Key details of the structural studies on GM in MS patients, including imaging technique, patient characteristics, depression/cognitive variables, and outcome.

| Reference | Imaging Technique | Subjects | Fatigue Scale | Matched Variables | Unmatched Variables | Neuroimaging Findings Correlated to Fatigue | Findings: F, NF vs. HC | Findings: F vs. NF |
|---|---|---|---|---|---|---|---|---|
| *Cross-sectional* | | | | | | | | |
| [53] | TBM | F:17 NF:17 | FSS | Disease duration, age, sex, immunomodulatory treatment, DSC score. | EDSS, central motor activation. | Atrophy: Mesial aspect of superior frontal gyrus R ($p = 0.027$), anterior cingulate, genual part R ($p = 0.030$); anterior insula and inferior frontal gyrus L ($p = 0.042$), inferior frontal gyrus L ($p = 0.004$), superior parietal lobule R ($p = 0.027$), inferior parietal lobule R ($p = 0.049$); inferior parietal lobule L ($p = 0.011$), middle temporal gyrus R ($p = 0.028$), superior temporal gyrus R ($p = 0.046$), caudate head R ($p = 0.039$) | | F↑ |
| [50] | MRI | F:71 NF81 | FSS | Sex, age, disease duration, T2 LV | EDSS | Volume of putamen ($p = 0.011$), caudatum ($p = 0.020$), and thalamus ($p = 0.004$). | | F↓ |
| | | | | | | Cth of the superior frontal gyrus ($p = 0.003$) and inferior parietal gyrus ($p = 0.001$) | | F↓ |
| | | | | | | Global Cth ($p < 0.001$), frontal lobe ($p < 0.001$), temporal lobe ($p < 0.001$) | F↓ | |
| | | | | | | Volume of putamen ($p < 0.001$), caudatum ($p < 0.001$), pallidus ($p < 0.001$), and thalamus ($p < 0.001$) | F↓ | |
| [78] | VBM | F:16 NF:13 | MFIS | Age, sex, education, disease duration | IFS, IC-AS | GM atrophy | | F = NF |
| | | | | | | GM volume interoceptive areas (thalamus, hippocampus, caudate R, putamen R, temporal mid R and L, temporal sup R and L, temporal pole sup R, cingulum mid L, cerebellum L and R, cuneus R, frontal sup orb L, frontal mid orb L and R, cingulum ant R, cingulum mid R and L, fusiform L) ($p < 0.001$) | F↓ | |
| | | | | | | GM volume (thalamus, hippocampus, vermis, cerebellum L, caudate R, putamen, frontal sup R, parahippocampal L, amygdala, precentral R, occipital mid R, putamen L, pallidum L, lingual L, occipital Mid L, postcentral L, cingulum Mmid L) ($p < 0.001$) | NF↓ | |
| [80] | VBM | F:21 NF:17 | MFIS | Age, sex, education, relationship status, EDSS, disease clinical phenotype, disease duration, pharmacological treatment | HADS, TAS | Volume of caudate nuclei R ($p = 0.011$), L ($p = 0.005$) | | F↑ |
| | | | | | | Volume of L parietal cortex ($p = 0.011$) | | F↓ |
| [99] | MT and DT MRI | F:14 NF:14 | FSS | Age, disease duration, EDSS | | Average MTR and MD from cerebral GM. GM of the frontal lobe's cerebral cortex and basal ganglia. | | F = NF F = NF |

**Table 3.** *Cont.*

| Reference | Imaging Technique | Subjects | Fatigue Scale | Matched Variables | Unmatched Variables | Neuroimaging Findings Correlated to Fatigue | Findings: F, NF vs. HC | Findings: F vs. NF |
|---|---|---|---|---|---|---|---|---|
| *Cross-sectional* | | | | | | | | |
| [59] | MRI | F:15 NF:12 | MFIS | Age, disease duration, annual relapse rate, EDSS, BDI, lesion relative fraction | | Thalamus volumes | | F = NF |
| | | | | | | Cth of Rolandic regions and the volume of thalami | | F = NF |
| [61] | VBM | F:32 NF:28 | FSS * | Sex, age, disease duration, T1 LV, ICV | EDSS, CDMI | GM volume: left cerebellum ($p < 0.001$). | | F↓ |
| | | | | | | GM atrophy in R paracentral gyrus (SMA), different areas of the bilateral temporal and occipital lobes, R precuneus, bilateral thalamus ($p < 0.001$) | NF↑ | |
| | | | | | | GM atrophy in the paracentral gyrus (SMA), bilateral precentral gyrus (PMC), bilateral occipital lobe, precuneus and posterior cingulate gyrus ($p < 0.001$) | F↑ | |
| [81] | MRI | F:22 NF:27 | FSS | Sex, age *, relapse in previous 24 months, disease duration, pharmacological treatment, PASAT | EQ5D, ZDS *, EDSS *, pyramidal FS score *, 9HPT, T25FW, SDMT Intracranial volume * | Atrophy of caudate (EDSS covariate: $p = 0.048$; depression covariate: $p = 0.046$), accumbens volumes (EDSS covariate: $p = 0.047$, depression covariate: $p = 0.042$), volume of cerebellar CLs (EDSS covariate: $p = 0.0099$, or pyramidal score: $p = 0.0002$) | | F↑ |
| [82] | VBM | F:11 NF:6 | EMIF-SEP | Age, sex, EDSS, disease duration, MADRS, Mattis score, lesion volume | | GM density in frontal mid L and frontal sup L ($p < 0.001$), frontal mid orb R ($p = 0.024$), frontal sup orb L, frontal med orb L and frontal mid orb L ($p = 0.007$), frontal inf tri L ($p = 0.008$), temporal inf L ($p < 0.001$), precuneus L and parietal sup L ($p < 0.001$). | | F↓ |
| [56] | VBM | F:64 NF:59 | FSS * | Sex, age, disease duration, pharmacological treatment, PASAT, T2 LV, T1 LV, NBV | EDSS, MADRS * | GM atrophy: thalamus, caudate nucleus, putamen, insula, amygdala, hippocampus, ACC, MCC, PCC, orbital SFG, orbital MFG, orbital IFG, IFG pars triangularis, IFG pars opercularis, medial SFG, SFG, MFG, SMA, paracentral lobule, precentral gyrus, postcentral gyrus, SPL, IPL, precuneus, cuneus, angular gyrus, Heschl gyrus, STG, ITG, MTG, fusiform gyrus, lingual gyrus, SOG, MOG, calcarine sulcus | | F = NF |
| [98] | MRI | F:18 NF:42 | FSS | Age, education, disease duration, EDSS, BPF, FSS, BDI, alertness without cueing, alertness with cueing, time walk test, 9-HPT, PASAT | BDI cognitive somatic items | Cth: right inferior parietal lobe ($p < 0.05$). | | F↓ |
| | | | | | | Cth: precuneus R ($p < 0.05$), middle cingulate R ($p < 0.05$) | F↓ | |
| [89] | MRI | F:174 NF:192 | MFIS | Sex, education, PASAT, disease duration | Age, EDSS, MADRS | Normalized brain volume, normalized GM volume, normalized thalamic volume | | F = NF |

**Table 3.** *Cont.*

| Reference | Imaging Technique | Subjects | Fatigue Scale | Matched Variables | Unmatched Variables | Neuroimaging Findings Correlated to Fatigue | Findings: F, NF vs. HC | Findings: F vs. NF |
|---|---|---|---|---|---|---|---|---|
| *Cross-sectional* | | | | | | | | |
| [52] | MRI | F:16 NF:17 | FSS | Age, disease duration, EDSS, 17-HDRS | | T2 for juxtacortical, periventricular, deep GM, infratentorial, deep WM. GM volume, WM volume, total brain volume | | F = NF |
| [83] | MRI | F:20 NF:11 | FSS | Age, sex, disease duration, T2 volume. | EDSS | Deep GM T1 in the thalamus ($p = 0.018$) | | F↑ |
| [84] | VBM | F:30 Reversible F:31 NF:37 | MFIS | Age, sex, disease duration, disease clinical phenotype, EDSS, timebetween MFIS and MRI | CES-D, WM LL | GM volume frontal pole, frontal gyrus, frontal-orbital cortex, frontal-medial cortex, cingulate gyrus, paracingulate gyrus, precentral gyrus, postcentral gyrus, insula, temporal pole, superior temporal gyrus, middle temporal gyrus, transverse temporal gyrus, planum temporale, planum polare, parahippocampal gyrus, precuneus, supramarginal gyrus, angular gyrus, lateral occipital cortex, hippocampus, amygdala, accumbens, caudate, putamen, thalamus, cuneus, occipital pole, periaqueductal GM, cerebellum (age, sex, disease duration, EDSS, CESD, medication family-wise error, Bonferroni corrected $p < 0.017$) | | F↓ |
| [85] | MRI | F:8 NF:16 | MFIS | NR | EDSS, CES-D *, age | CTh parietal lobe ($p = 0.05$) Thalamic volume ($p = 0.07$) | | F↓ |
| [40] | MRI | F:10 NF:14 | FSS | Sex, age, disease duration, EDSS, T2LV, NBV, WMV, GMV. | | GM atrophy L central culcus, L middle frontal gyrus, precentral gyrus ($p < 0.05$, family-wise error corrected) | | F↑ |
| | | | | | | GM atrophy: L superior frontal sulcus, L precentral gyrus, posterior cingulate cortex, R thalamus, L middle frontal gyrus ($p < 0.05$; family-wise error corrected) GM atrophy: L central sulcus, L middle frontal gyrus ($p < 0.05$; family wise error corrected) | F and NF↑ F↑ | |
| [76] | DT MR | F: 31 NF:32 | FSS | Sex, age, disease duration, EDSS, disease clinical phenotype, pharmacological treatment, MADRS, T2 LV, T1 LV. | | Atrophy of R side of the nucleus accumbens ($p = 0.01$) GM atrophy R ITG (BA20) ($p < 0.001$, uncorrected), | | F↑ |
| | | | | | | GM atrophy in R thalamus, L side of the hippocampus, L side of the caudate nucleus, R inferior frontal gyrus, R middle temporal gyrus, R middle cingulate gyrus, L superior frontal gyrus, R ITG, L middle frontal gyrus, R anterior cingulate gyrus ($p < 0.001$, uncorrected) R thalamus, L thalamus, R postcentral gyrus, L caudate nucleus ($p < 0.001$ uncorrected) | F↑ NF↑ | |

**Table 3.** *Cont.*

| Reference | Imaging Technique | Subjects | Fatigue Scale | Matched Variables | Unmatched Variables | Neuroimaging Findings Correlated to Fatigue | Findings: F, NF vs. HC | Findings: F vs. NF |
|---|---|---|---|---|---|---|---|---|
| *Cross-sectional* | | | | | | | | |
| [86] | MRI, VBM | F:43 NF:17 | MFIS | | T2 LL, T1 LL. | GM atrophy in the left superior frontal gyrus ($p = 0.006$), R middle frontal gyrus ($p = 0.008$), and L middle frontal gyrus ($p = 0.009$) | | F↑ |
| | | | | | | GM atrophy in the left superior frontal gyrus ($p < 0.001$), R middle frontal gyrus ($p < 0.001$), and L middle frontal gyrus ($p < 0.001$) | F and NF↑ | |
| [87] | MRI | F:197 NF:25 | FSS | Age at onset, number of relapses, WM-f. | Age, disease duration, education, AWM-f, GM-f, T2 lesion, T1 lesion. | GM-f ($p < 0.001$) | | F↓ |
| [51] | MRI | F:11 NF:9 | MFIS | Age, sex, disease duration, relapse, EDSS, FSS, BDI, 9-HPT | | Global Cth | | F = NF |
| [88] | MRI | F:23 NF:9 | FSS | Sex, age, disease duration, T2 LV | EDSS | Hypothalamic volume | | F = NF |

\* covariate. **Legend**. ACC: anterior cingulate cortex; AWM: abnormal white matter fraction BDI: Beck depression inventory; BPF: brain parenchymal fraction; CDMI: Chicago multiscale depression inventory; CES-D: Center for Epidemiologic Studies depression scale; Cth: cortical thickness; DT: diffusion tensor; DSC: digit symbol coding; EDSS: expanded disability status scale; EMIF-SEP: French version of fatigue impact scale; F: patients with fatigue; FS: functional scale; FSS: fatigue severity scale; GM: gray matter; HADS: hospital anxiety and depression scale; 9-HPT: 9-hole peg test; 17-HDRS: Hamilton depression rating scale; IC-AS: interoceptive condition-accuracy score; ICV: intracranial volume; IFG: inferior frontal gyrus; IFS: INECO frontal screening; IPL: inferior parietal lobule; ITG: inferior temporal gyrus; L: left; LL: lesion load; LV: lesion volume; MADRS: Montgomery Asberg depression rating scale; MCC: middle cingulate cortex; MFG: middle frontal gyrus; MFIS: modified fatigue impact scale; MOG: middle occipital gyrus; MRI: magnetic resonance imaging; MT: magnetization transfer; MTG: middle temporal gyrus; MTR: magnetization transfer ratio; NBV: normal brain volume; NF: patients without fatigue; NR: not reported; PASAT: paced auditory serial addition test; PCC: posterior cingulate cortex; R: right; SDMT: symbol digit modalities test; SFG: superior frontal gyrus; SMA: supplementary motor area; SOG: superior occipital gyrus; SPL: superior parietal lobule; STG: superior temporal gyrus; TAS: Toronto alexithymia scale; TBM: tensor based morphometry; T25FW: timed 25-foot walk test; VBM: voxel-based morphometry; WM: white matter; ZDS: Zung self-rating depression scale.

**Table 4.** Key details of the structural studies on GM in MS patients with cognitive fatigue, including imaging technique, patient characteristics, depression/cognitive variables, and outcome.

| Reference | Imaging Technique | Subjects | Fatigue Scale | Matched Variables | Unmatched Variables | Neuroimaging Findings Correlated to Fatigue | Findings: F, NF vs. HC | Findings F vs. NF |
|---|---|---|---|---|---|---|---|---|
| *Cross-sectional* | | | | | | | | |
| [89] | MRI | CF: 115 CNF: 251 | MFIS | PASAT, disease duration, EDSS | Sex, age, education, MADRS | Normalized brain volume, normalized GM volume, normalized thalamic volume | | CF = CNF |

Legend. CF: patients with cognitive fatigue, CNF: patients without cognitive fatigue, EDSS: expanded disability status scale, GM: gray matter, MADRS: Montgomery Asberg depression rating scale, MFIS: modified fatigue impact scale; MRI: magnetic resonance imaging, PASAT: paced auditory serial addition test.

**Table 5.** Key details of the functional studies in MS patients, including imaging technique, patient characteristics, depression/cognitive variables, and outcome.

| Reference | Imaging Technique | Subjects | Fatigue Scale | Matched Variables | Unmatched Variables | Neuroimaging Findings Correlated to Fatigue | Findings: F, NF vs. HC | Findings F vs. NF |
|---|---|---|---|---|---|---|---|---|
| *Cross-sectional* | | | | | | | | |
| [60] | rs-fMRI | F:28, NF:31 | FSS | Age, sex, disease duration, education, EDSS, PASAT, T2LV, NBV | NR | DMN FC in the PCC ($p < 0.05$) | F and NF↓ | F↑ |
| | | | | | | DMN FC in ACC ($p < 0.05$). SMN FC activation in the L PMC and SMC SMN FC in R PMC, L PMC ($p < 0.05$) | F↓ NF↑ | F↓ F↑ |
| [78] | rs-fMRI | F:16 NF:13 | MFIS | Age, sex, education, disease duration | IFS, IC-AS | FC between R ACC and L insula ($p = 0.002$) | F↑ | |
| [61] | rs-fMRI | F:32 NF:28 | FSS * | Sex, age, disease duration, T1 LV, ICV | EDSS, CDMI | SMN: rs-FC: left precentral gyrus associated with premotor cortex ($p < 0.005$, family-wise error corrected) SMN: rs-FC of the R precentral gyrus and PMC ($p < 0.005$, family-wise error corrected) | NF↑ | NF↑ |
| [82] | VBM, PET | F:11 NF:6 | EMIF-SEP | Age, sex, EDSS; disease duration, MADRS, Mattis score, lesion volume | NR | rCMRglu | | F = NF |
| [62] | Task-based fMRI (Hand motor task) | F:15 NF:14 | FSS | Age, disease duration, EDSS. | NR | Relative activation of the contralateral CMA ($p = 0.001$) | | F↑ |
| | | | | | | Activation of ipsilateral cerebellar hemisphere ($p = 0.004$), the ipsilateral rolandic operculum ($p = 0.001$), the ipsilateral precuneus ($p < 0.001$), the contralateral thalamus ($p < 0.001$), and the contralateral middle frontal gyrus ($p = 0.003$) Activation of ipsilateral inferior frontal gyrus ($p = 0.01$) and contralateral thalamus ($p = 0.001$) | F↓ | NF↑ |
| [63] | rs-fMRI | F:36 NF:86 | MFIS | Sex, pharmacological treatment | Age, education, disease clinical phenotype, EDSS, T2 LV, T1 LV, NBV | rs-FC between L temporal SR and cerebellum ($p < 0.05$, family-wise error corrected) | | F↑ |
| | | | | | | rs-FC between L motor SR and insula ($p < 0.05$ family-wise error corrected), L temporal SR and cerebellum ($p < 0.05$ family-wise error corrected) | NF↑ | |
| [64] | Task-based fMRI (repetitive flex-ext of the last four fingers of the right hand moving together) | F:50 NF:29 | MFIS | Sex, age, disease duration, EDSS, T2 LV, T1 LV | NR | Activation of bilateral MTG, left pre-SMA, left SMA, bilateral superior frontal gyrus, left postcentral gyrus, left putamen, and bilateral caudate nucleus ($p < 0.05$ family-wise error corrected). | F↓ | F↓ |
| | | | | | | Activation in R middle frontal gyrus ($p < 0.05$ family-wise error corrected), Activation of R precentral gyrus, R middle temporal gyrus, and bilateral cerebellum ($p < 0.01$) | F↑ F and NF↑ | F↑ |

**Table 5.** *Cont.*

| Reference | Imaging Technique | Subjects | Fatigue Scale | Matched Variables | Unmatched Variables | Neuroimaging Findings Correlated to Fatigue | Findings: F, NF vs. HC | Findings F vs. NF |
|---|---|---|---|---|---|---|---|---|
| *Cross-sectional* | | | | | | | | |
| [65] | Task-based fMRI (Task1: flex-ext of the last four fingers of the hand. Task2: flex-ext of the hand and foot in phasic) | F:12 NF:10 | FSS | Age, disease duration, EDSS, 9-HPT, finger and foot tapping rate, pharmacological treatment | NR | Task 1: Recruitment of ipsilateral thalamus, contralateral CMA, regions located in the MFG, bilaterally. Primary SMC bilaterally, SMA bilaterally ($p < 0.05$ corrected for multiple comparison) | | F↑ |
| | | | | | | Task 2: Activation of the thalamus bilaterally, contralateral primary SMC, and contralateral precentral gyrus ($p < 0.05$ corrected for multiple comparison). | | F↑ |
| | | | | | | Activation of the contralateral SII ($p < 0.05$ corrected for multiple comparison). | | NF↑ |
| [66] | Task-based fMRI (cycle movement of the hand and foot) | F:11 NF:13 | FSS | Sex, age, disease duration, EDSS | NR | In-phase movement: activation cerebellum bilaterally, R precuneus, R MFG, SMA bilaterally, L hand primary SMC ($p < 0.05$ corrected at a cluster-level) | NF↑ | |
| | | | | | | In-phase movement: activation cerebellum bilaterally, L SII, R precuneus, L hand primary SMC ($p < 0.05$ corrected at a cluster-level) | F↑ | |
| | | | | | | In-phase movement: activation L cerebellum, L SII ($p < 0.05$ corrected at a cluster-level) | | F↑ |
| | | | | | | Anti-phase movement: activation L cerebellum, L SII, R precuneus, L IPL, R MFG, L MFG, L IFG, B CMA, B SMA, L hand primary SMC ($p < 0.05$ corrected at a cluster-level) | NF↑ | |
| | | | | | | Anti-phase movement: activation cerebellum bilaterally, L SII, R precuneus, L hand primary SMC ($p < 0.05$ corrected at a cluster-level) | F↑ | |
| | | | | | | Anti-phase movement: activation cerebellum bilaterally, L SII, R precuneus ($p < 0.05$ corrected at a cluster-level) | | F↑ |
| [91] | Task-based fMRI (tactile stimulation of the palm of the right hand) | F:20 NF:15 | FSS | Sex, age, EDSS, disease duration | NR | Cervical cord mean fMRI intensity ($p = 0.04$) Cervical cord mean fMRI intensity ($p = 0.02$) | NF↑ | NF↑ |
| [67] | PET | F:19 NF:16 | FSS | Age at onset of MS symptoms, age at PET investigation, disease duration, EDSS | NR | CMRGlu bilaterally in a prefrontal lobe including the lateral and medial prefrontal cortex and adjacent WM, in the premotor cortex, and in the right SMA area. Capsula interna and extended from the ventral putamen toward the lateral head of the caudate nucleus, particularly at the R brain side. Posterior parietal cortex ($p < 0.005$) (Brodman area [BA] 39/40, supramarginal and angular gyrus, medial occipital gyrus), which extended into the middle temporal and occipital gyrus ($p < 0.005$). | | F↓ |
| | | | | | | R cerebellar vermis and to the anterior cingulate gyrus of both brain sides | | F↑ |
| | | | | | | Global CMRGlu ($p = 0.0014$) Global CMRGlu ($p = 0.0008$) | F↓ NF↓ | |

**Table 5.** *Cont.*

| Reference | Imaging Technique | Subjects | Fatigue Scale | Matched Variables | Unmatched Variables | Neuroimaging Findings Correlated to Fatigue | Findings: F, NF vs. HC | Findings F vs. NF |
|---|---|---|---|---|---|---|---|---|
| *Cross-sectional* | | | | | | | | |
| [68] | Task-based fMRI (finger tapping) | F:12 NF:12 | FSS | Age, sex, hand dominance, depression, clinical disability, disease duration, motor performance | NR | Activation of the premotor area ipsilateral* at the level of the R putamen ($p = 4.26$) and of the middle frontal gyrus ($p = 3.30$) on the R DLPFC ($p = 3.12$). Bilateral activation of the SMA and ipsilateral activation of the premotor cortex and cerebellum. | | F↑ |
| | | | | | | Activation of primary sensorimotor areas bilaterally (R: $p = 3.34$), R SMA ipsilateral ** ($p = 4.27$), L premotor area contralateral ** ($p = 3.46$), cerebellum contralateral ** ($p = 3.56$), upper parietal lobe bilaterally (R: $p = 3.88$; L: $p = 3.60$). | NF↑ | |
| [69] | rs-fMRI | F:10 NF:12 | FSS | Age, disease duration, LL, LV | MFIS, BDI | Connectivity between the R thalamus and R precentral gyrus ($p = 0.015$). | | F↑ |
| | | | | | | Connectivity between R thalamus and L parietal operculum ($p = 0.0002$), L thalamus and R superior frontal gyrus ($p = 0.046$), and between the L insula and posterior cingulate ($p = 0.003$). | | F↓ |
| [92] | Task-based fMRI (pincer grip, produced a steady force level: 20% MVC) | F:27 NF:17 | FSMC | Age, gender, disease duration, treatment, PSQI, ESS, PASAT, SDMT, JTHFT, 9-HPT | EDSS, BDI | Task-related activity pattern | F and NF = HC | F = NF |
| [93] | MRSI | F:34 NF:26 | FSS | EDSS, Age, disease duration, T2 LV, FSS | NR | The NAA/Cr ratio (controlling for EDSS and age, $p = 0.004$) | F↓ | |
| [94] | MRSI | F:17 NF:13 | FSS, MFIS | Age, sex, disease duration, | EDSS *, BDI * lesion volume * | NAA/Cr in the lentiform nucleus region (Controlling for LV, BDI, and EDSS, $p = 0.015$) | F↓ | |
| [95] | MRSI | F:10 NF:9 | FSS | Age *, EDSS, LL | %GM * | In the pons, NAA/tCr in L4, R5 and R6 | F↓ | |
| | | | | | | In the pons, NAA/tCr in L6 | NF↓ | |

* covariate ** to the movement. **Legend.** ACC: anterior cingulate cortex; BDI: Beck depression inventory; CDMI: Chicago multiscale depression inventory; CMA: cingulate motor area; DMN: default mode network; DLPFC: dorsolateral prefrontal cortex; EDSS: expanded disability status scale; EMIF-SEP: French version of fatigue impact scale; ESS: Epworth sleepiness scale; F: patients with fatigue; FC: functional connectivity; fMRI: functional magnetic resonance imaging; FSMC: fatigue scale for motor and cognitive function; FSS: fatigue severity scale; GM: gray matter; HC: healthy control; 9-HPT: 9-hole peg test; IC-AS: interoceptive condition-accuracy score; ICV: intracranial volume; IFG: inferior frontal gyrus; IFS: INECO frontal screening; IPL: inferior parietal lobule; JTHFT: Jebsen Taylor hand function test; LL: lesion load; MADRS: Montgomery Asberg depression rating scale; MFG: middle frontal gyrus; MFIS: modified fatigue impact scale; MRI: magnetic resonance imaging; MRSI: magnetic resonance spectroscopic imaging; MTG: middle temporal gyrus; MVC: maximal voluntary contraction; NAA/Cr: N-acetylaspartate/creatine; NBV: normal brain volume; NF: patients without fatigue; NR: not reported; PASAT: paced auditory serial addition test; PET: positron emission tomography; PCC: posterior cingulate cortex; PMC: primary motor cortex; PSQI: Pittsburgh sleep quality index; R: right; rCMRglu: relative glucose metabolism; rs-FC: resting-state functional connectivity; rs-fMRI: resting-state fMRI; SDMT: symbol digit modalities test; SMA: supplementary motor area; SII: secondary sensorimotor cortex; SMC: sensorimotor cortex; SMN: sensory motor network; T1LV: T1 lesion volume T2LV: T2 lesion volume; VBM: voxel-based morphometry; WM: white matter.

**Table 6.** Key details of the functional studies in MS patients with cognitive fatigue, including imaging technique, patient characteristics, depression/cognitive variables, and outcome.

| Reference | Imaging Technique | Subjects | Fatigue Scale | Cognitive Evaluation | Matched Variables | Unmatched Variables | Neuroimaging Findings Correlated to Fatigue | Findings: CF, CNF vs. HC | Findings CF vs. CNF |
|---|---|---|---|---|---|---|---|---|---|
| *Cross Sectional* | | | | | | | | | |
| [97] | Task-based fMRI (paced auditory serial addition test (PASAT)) | CF:11 CNF:11 | FSMC | PASAT: CF:81.2(47–118) CNF:103.6(73–118) | Age, sex, education, disease duration, EDSS, NBV, NGMV, NWMV, T2LV | NR | RS-FC at t2 (30 min after execution of PASAT) between the L superior frontal gyrus and supplementary motor area, bilateral middle temporal gyri and the bilateral middle occipital gyri ($p < 0.001$, uncorrected), the L-superior frontal gyrus (SFG) hyperconnected at t1(immediately after PASAT) with the left caudate nucleus and hypoconnected at t2 with the left anterior thalamus. | CF↑ | CF↑ |

**Legend**. BDI: Beck depression inventory; CF: patients with cognitive fatigue; CNF: patients without cognitive fatigue; EDSS: expanded disability status scale; fMRI: functional magnetic resonance imaging; FSMC: fatigue scale for motor and cognitive function; NBV: normal brain volume; NGMV: normal gray matter volume; NWMV: normal white matter volume; rs-FC: resting-state functional connectivity; PASAT: paced auditory serial addition test; T2LV: T2 lesion volume.

Rs-fMRI (BOLD): Comparing F vs NF patients, one study showed a higher default mode network (DMN) FC in the posterior cingulate cortex (PCC) and a lower one in the anterior cingulate cortex (ACC) in F compared to NF patients [60]. Three studies reported that the sensorimotor network (SMN) FC resulted higher in F compared to NF patients [60,63,69]. On the other hand, two papers found lower FC in F compared to NF patients between subcortical regions. [63,69]. Resting-state FC resulted higher in NF compared to F patients between left precentral gyrus and premotor cortex [61]. Only one study did not find any difference in terms of FC between F and NF patients in whole brain [96]. Rs-FC resulted lower in DMN in ACC in F patients compared to HC; on the other hand, PCC resulted higher in F and NF patients compared to HC. Considering SMN, FC resulted higher in F and NF patients compared to HC [60,61,64]. Moreover, two papers reported a significant difference in terms of rs-FC in NF patients compared to HC [63] and F patients compared to HC [96].

Resting-state brain perfusion, metabolism, and metabolites: One paper showed reduced cerebral glucose metabolism in F compared to NF patients [67]. Additionally, three papers reported the NAA/Cr ratio reduced in F patients [93–95]. One study did not find any differences between F and NF patients in terms of relative glucose metabolism (rCMRglu) [82] and in terms of choline/creatine ratio (Cho/Cr) [93]. Global CMRGlu and ratio of N-acetylaspartate to total creatine (NAA/tCr) resulted lower in F and NF patients compared to HC [67,95].

**Task-based fMRI:**

Using block scan design (ABAB), in three studies F and NF patients were scanned while performing a simple finger task: finger tapping [65,68] and finger flex-extension [62]. F patients showed a higher activation of cortical and subcortical areas than NF patients [62,65,68]. One study reported a lower fMRI activity occurrence in C5 and C6 during a tactile stimulation of the palm of the right hand [91]. One study just reported the scanning results before the fatiguing task (tonic grip force), and they showed a higher activation of left dorsal premotor cortex and prefrontal cortex rostral to the pre-supplementary area in NF than in F patients [92]. On the other hand, in another study they reported higher activation in F than NF patients using coordinated hand and foot movements [66].

Using task-based fMRI, one study reported rs-FC at t0 (immediately before paced auditory serial addition task (PASAT)), t1 (immediately after PASAT), t2 (30 min after execution of PASAT). The most relevant results were a higher rs-FC at t2 in CF compared to CNF [97]. During a hand motor task, F patients presented a lower activation of cortical and subcortical regions [62,64] compared to HC. Moreover, NF patients reported a higher activation of motor areas compared to HC [68].

## 4. Discussion

The main aim of the present study was to describe potential correlations between brain structural and functional alteration and symptoms of fatigue in patients affected by MS. We presented results of fifty studies. Structural and functional findings will be discussed separately.

### 4.1. Structural Analysis

**Conventional MRI and atrophy**: During the last decade, conventional sequences, such as FLAIR (fluid attenuated inversion recovery), T2-weighted sequences, and gadolinium-enhanced T1-weighted sequences, have been recognized as the most sensitive and reproducible methods of damage identification due to MS-like plaques, inflammatory activity, and LL [100]. In the last few years, non-conventional MR-derived metrics for brain imaging have been developed. They can be used to quantify relevant features of MS pathology and to observe the reparative mechanisms. These metrics include: measures of hypointense T1 lesions, CNS atrophy, and MTR [100]. Indeed, the development of automated techniques, such as VBM or FreeSurfer, to analyze structural MRI data allows one to study focal differences in brain anatomy that sometimes are not perceptible by visual inspection [101].

The measurement of brain MRI LL in MS has allowed the definition of clinical/MRI correlations, the natural course of the disease, and the efficacy of treatment [102]. Part of the total WM damage is shown by the T2 hyperintense lesions. These lesions reveal focal demyelination and axonal loss [103]. To perform an analogous activity, the destruction of the axons in the CNS leads to recruitment of more nerve fibers or areas in the brain in patients with MS compared to healthy people. This may exacerbate the phenomenon of fatigue [62]. Based on this hypothesis, most of the studies evaluate the relationship between the lesion status and the symptom of fatigue in patients. Several studies assessed the differences between F and NF patients in terms of LL, LV, WM, and GM atrophy, normal-appearing white matter (NWMV), and normal-appearing gray matter (NGMV). Since the different outcomes denote different concepts, the results of all of these studies make the comparison challenging.

In contrast with the hypothesis that most of the studies made, the majority of them did not find a significant difference between F and NF patients [52,57,60,94,104], even when they considered the global brain [76,82]. Only in frontal and temporal areas does there seem to be evidence of different lesion occurrence between F and NF patients [39,76,82,87]. The results may be influenced by the level of disability of patients included in the studies. Most of the studies matched the patients for disease duration, EDSS, and disease clinical phenotype. It is well known that the score of EDSS is correlated with the LL [105], and a sample with high level of disability could have precluded the identification of LL differences between patient groups. Since the MS patients usually present the symptom of fatigue concomitant to other symptoms (such as depression, pain, etc.), the small sample size of pure fatigue patients might influence the LL results.

It is important to highlight that in patients affected by MS, the decrease of brain volume has been correlated with disability progression and cognitive impairment. Specifically, the loss of GM volume is more nearly correlated with clinical impairment than a loss of WM volume [106]. Several MRI-based methods have been utilized for the assessment of *global or regional brain volume*, including cross-sectional and longitudinal techniques. One of the most important cross-sectional methods utilized in numerous studies is the automated technique: VBM. VBM is based on the voxel-wise comparison of the regional volume or concentration of GM and WM between subject groups [101]. Several studies reported the correlation between fatigue and brain atrophy [53,87,107]. Specifically, it seems that subcortical regions, in particular thalamus and prefrontal cortex, are the most involved area [50,53,59,80,81,96]. According to the literature, dysfunction in the thalamus seems to be related with fatigue in patients with MS [108]. Indeed, it is important to note that A Chaudhuri and PO Behan [109] associated "central fatigue" with structural damage in the component of the fronto-striato-thalamic circuits. They hypothesized that fatigue might be caused by a disparity in perception of energetic costs of an action (effort) and benefits of the consequent outcome (reward). Conventionally, the fronto-striato-thalamic circuits can be divided into sensorimotor, associative, and limbic loops [110], but recently it has been demonstrated that there is an intricate interplay between these loops and other brain structures outside these circuits which combine different components of reward mechanisms: reward evaluation, associative learning, the capacity to formulate appropriate action plans and inhibit inappropriate choices based on earlier experience [58].

Moreover, GM atrophy in the basal ganglia and the limbic system seems to be common to symptoms of fatigue, pain, and depression [16]. It is important to note that it has been demonstrated that the prefrontal cortex contributes to the top-down regulation of sensory and affective processes, and its projections to the periaqueductal gray, thalamus, and amygdala have been demonstrated to influence chronic pain phenotypes [111,112].

Zhou et al. [113] showed that inhibiting these pathways worsens pain, demonstrating that this pathway is employed endogenously to suppress pain. The structural alteration in prefrontal cortex reported in patients with fatigue might be related to a major sensitivity to pain that enhances the symptom of fatigue.

On the other hand, only four studies did not report any differences between two groups of patients [52,56,96,99]. The different phenotype of MS (RRMS, PPMS, SPMS) included in the sample of participants might have influenced the results. It has been demonstrated that the progression of GM atrophy is not the same across the stages of MS [114]. Moreover, the matched variables between groups were not the same for all the studies. It is important to note that, other symptoms as depression, is correlated with fatigue [115]. Since that, the matched between groups should consider all this factor that could affect the fatigue in patients with MS.

Another factor of impact on GM volume reduction is Cth. FreeSurfer Image Analysis Suite is a software that estimates Cth by calculating the distance between WM margin and cortex [101]. It has been used to examine the difference between F and NF patients in two studies, which both reported that the mean global Cth is not different in F and NF patients [50,51]. It is important to note that when Bonferroni's correction is applied in the region of interest (ROI), F patients significantly differed from NF patients in the Cth, specifically in the superior frontal gyrus, inferior parietal gyrus [50], and in the parietal lobe [85]. It is important to note that the global Cth is significantly different between F patients and HC [50]. Moreover, the same measure was obtained using "MeVisCTM", a semi-automatized application of NeuroQLab3.531, in one study which found a significant decrease of Cth in F compared to NF patients and HC, only in the inferior parietal lobe [98].

*DWI:* DWI are based on the assessment of water molecules' motion within the tissue, and the alteration of brain structure caused by MS might affect water motion [116]. DWI may provide information on WM and GM architecture and the integrity of MS patients' brains [117]. Moreover, they could indicate the brain microstructural damage outside of the focal WM lesions, in the NAWM and in the NAGM [118]. The evidence supplied the relationship between diffusion abnormalities and the clinical condition of patients affected by MS; alteration of DWI values is more significant in patients with severe EDSS and with long disease [119–121].

DWI techniques provide important indices in order to evaluate the integrity of WM. First of all, FA gives information on the degree of diffusion directionality and ranges from 0 (isotropic diffusion) to 1 (anisotropic diffusion). FA diminished in focal WM lesions typical in patients with MS [122,123]. Moreover, the diffusion rate along the principal axis of diffusion (AD), the molecular diffusion rate (MD), and the rate of diffusion in the transverse direction (RD) allow us to make hypotheses within voxel about tissue proprieties [101]. Using DWI, most of the studies suggested an association between diffusion alterations and fatigue [55,70,75–77,98]. Specifically, FA seems to be reduced in F compared to NF patients [55,75–77] and to HC [75,124]. Since FA reveals focal lesions in WM and in NAWM, it is considered a sensitive tool [122]. The differences in terms of FA between F and NF patients may indicate a correlation between the coherence of WM and the symptom of fatigue in patients with MS.

The structural results could be influenced by the evaluation of fatigue in patients. All structural studies reported the cutoff scale in order to separate patients' groups, but they were not the same for all reported studies. An objective and standard method is required in order to evaluate the level of fatigue in patients with MS.

### 4.2. Functional Results

Despite the fact that structural neuroimaging plays a key role in the diagnosis of MS, the use of functional imaging is a rather new area of research. Neuroimaging measures collected in the review include fMRI, resting-state MRSI, and PET. The first one detects the brain's active part by assessing the changes in terms of blood oxygen level-dependent (BLOD) [125,126]. MRSI allows us to evaluate the metabolism and metabolites of the tissue. PET affords a means to image and measure biological processes' rates across the distributed and interrelated systems of the brain [127].

The functional analysis reported regional differences between F and NF patients. Using fMRI, in resting state, one of the most significant results was FC detected in SMN

or in DMN [60,61]. Specifically, FC in premotor cortex and supplementary motor cortex (SMC) resulted higher in F than NF patients and HC [60]. However, rs-FC in precentral gyrus and premotor cortex was higher in NF compared to F patients and HC [61]. Since it has been demonstrated that FC changes appear corresponding to the clinical condition of patients [128], the unequal sample in terms of level of disability may have impacted results in terms of FC. Indeed, considering patients with fatigue, depression, and pain, functional changes were found in prefrontal cortex, basal ganglia, and limbic system results, crucial structures in valence and reward processing [16].

When participants were asked to perform a simple task during fMRI, three studies agreed about a significantly lower activation of cingulate motor area (CMA), ipsilateral supplementary motor area (SMA), and contralateral primary motor cortex (PMC) in F compared to NF patients [62,66,68]. Considering the sub-domain of cognitive fatigue, higher rs-FC in CF was found between superior frontal gyrus and occipital and temporal areas after PASAT [97].

In terms of the cerebral metabolic rate of glucose (CMRGlu), one study suggested a reduction of CMRGlu in the bilateral PMC and SMA in F compared to NF patients affected by MS [67]. It is well known that the motor cortex is involved in the planning, control, and execution of voluntary movement, and each motor cortex area has a different role in sequential motor control [129]. Specifically, SMA's function includes the internal generation of movement, bimanual coordination, and regulation of posture [130]. The increased and decreased inhibition of the sensorimotor network may play a role in the development of fatigue in patients affected by MS.

Biochemical changes in patients with MS were reported in relation to NAA/Cr evaluation. F patients showed a significantly lower NAA/Cr ratio compared to NF patients, which suggests a higher neuronal damage in F than NF patients [93–95].

### 5. Conclusions

Although evidence suggests a correlation between fatigue and thalamus/sensorimotor network dysfunction, the variability in terms of paradigm design, data acquisition, and analysis methods does not allow us to determine the exact mechanism underling the development of fatigue in patients with MS. Future research is necessary in order to better understand the correlation of fatigue and structural/functional alteration. Moreover, since the fatigue in patients with MS is influenced by other symptoms, such as depression and pain, or pharmacological treatment and autonomic nervous system imbalance, the future studies should consider a multiparametric approach.

**Supplementary Materials:** The following supporting information can be downloaded at: https://www.mdpi.com/article/10.3390/neurolint14020042/s1, Table S1: Demographics and clinical data of included studies.

**Author Contributions:** C.B., M.V., S.T. and A.M. conceived of the presented idea. C.B. and A.M. developed the literature analysis following the PRISMA method. S.T. and F.B.P. verified the methods and discussed with C.B. and A.M. about the results. A.P., F.G.L., G.G., C.M. and F.S. encouraged C.B. to investigate fatigue in multiple sclerosis and supervised the findings of this work. All authors have read and agreed to the published version of the manuscript.

**Funding:** This research received no external funding.

**Institutional Review Board Statement:** Not applicable.

**Informed Consent Statement:** Not applicable.

**Data Availability Statement:** Not applicable.

**Conflicts of Interest:** The authors declare no conflict of interest.

## Abbreviations

| | |
|---|---|
| MS | Multiple sclerosis |
| PCC | Posterior cingulate cortex |
| CNS | Central nervous system |
| ACC | Anterior cingulate cortex |
| MRI | Magnetic resonance imaging |
| SMN | Sensorimotor network |
| fMRI | Functional magnetic resonance imaging |
| rCMRglu | Relative glucose merabolism |
| PET | Positron emission tomography |
| Cho/Cr | Choline/creatine ratio |
| F | Patients with fatigue |
| NF | Patients without fatigue |
| CF | Patients with cognitive fatigue |
| CNF | Patients without cognitive fatigue |
| RRMS | Relapsing-remitting |
| PPMS | Primary progressive |
| SPMS | Secondary progressive |
| HC | Healthy control |
| EDSS | Expanded disability status scale |
| WM | White matter |
| GM | Gray matter |
| LL | Lesion load |
| SPM | Statistical parametric mapping |
| LV | Lesion volume |
| NAA/tCr | N-acetylaspartate to the total creatine |
| PASAT | Paced auditory serial addition task |
| NAWM | Normal-appearing white matter |
| NAGM | Normal-appearing gray matter |
| SMA | Supplementary motor area |
| PMC | Primary motor cortex |
| CMA | Cingulate motor area |
| VBM | Voxel-based morphometry |
| Cth | Cortical thickness |
| DWIs | Diffusion-weighted images |
| DTI | Diffusion tensor imaging |
| RD | Radial diffusivity |
| MD | Mean diffusivity |
| AD | Axial diffusivity |
| MTR | Magnetization transfer ratio |
| rs- fMRI | Resting-state fMRI |
| FC | Functional connectivity |
| NAA | N-acetylaspartate |
| Cr | Creatine |
| MRSI | Proton MR spectroscopic imaging |
| DMN | Default mode network |

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
