# Peer review of "Brain Structural and Functional Alterations in Multiple Sclerosis-Related Fatigue: A Systematic Review"

_2035-8377, doi:10.3390/neurolint14020042_

Round 1

Reviewer 1 Report

The article is well written, methods are complete and results detailed. I will just add some pictures as an example of brain structural and functional alterations in MS patients with fatigue. 

Moreover, I suggest to add the following references: 

  • Manjaly ZM, Harrison NA, Critchley HD, Do CT, Stefanics G, Wenderoth N, Lutterotti A, Müller A, Stephan KE. Pathophysiological and cognitive mechanisms of fatigue in multiple sclerosis. J Neurol Neurosurg Psychiatry. 2019 Jun;90(6):642-651. doi: 10.1136/jnnp-2018-320050. Epub 2019 Jan 25. PMID: 30683707; PMCID: PMC6581095.
  • Rottoli M, La Gioia S, Frigeni B, Barcella V. Pathophysiology, assessment and management of multiple sclerosis fatigue: an update. Expert Rev Neurother. 2017 Apr;17(4):373-379. doi: 10.1080/14737175.2017.1247695. Epub 2016 Oct 21. PMID: 27728987.

Author Response

The article is well written, methods are complete and results detailed.

We thank you for reviewing our manuscript and for your supportive words. We have considered your comments and made changes in the revised manuscript. We appreciate your further perusal of the revised manuscript.

To facilitate the identification of the revisions, we have highlighted in red all new or modified sentences.

I will just add some pictures as an example of brain structural and functional alterations in MS patients with fatigue. 

ANSWER: We thank the reviewer for this suggestion, and we added Figure 2 and Figure 3 in the appropriate space of the R1 version of the manuscript (3.2. Figures, Tables and Schemes).

Moreover, I suggest to add the following references:

Manjaly ZM, Harrison NA, Critchley HD, Do CT, Stefanics G, Wenderoth N, Lutterotti A, Müller A, Stephan KE. Pathophysiological and cognitive mechanisms of fatigue in multiple sclerosis. J Neurol Neurosurg Psychiatry. 2019 Jun;90(6):642-651. doi: 10.1136/jnnp-2018-320050. Epub 2019 Jan 25. PMID: 30683707; PMCID: PMC6581095.

Rottoli M, La Gioia S, Frigeni B, Barcella V. Pathophysiology, assessment and management of multiple sclerosis fatigue: an update. Expert Rev Neurother. 2017 Apr;17(4):373-379. doi: 10.1080/14737175.2017.1247695. Epub 2016 Oct 21. PMID: 27728987.

ANSWER: We thank the reviewer for this suggestion and we added in the introduction part the two references.

“It seems that the peripheral and central immunological and inflammatory process might play a central role in the exacerbation of fatigue, specifically in patients with MS [1].

Recently, some studies have provided potential mechanisms that underlying the subjective experience of fatigue [2-5], such as metacognitive mechanisms [1].

Moreover, in clinical practise the use of reliable and standardize fatigue scale is essential to plan and supervise an adequate personalized treatment strategy [6]. But, the large scale heterogeneity and a missed consensus on management of fatigue make challenging the control of this symptom in patients with MS [6].”

Namely, patients with fatigue reported an increase of distributed brain activity during the performance of tasks [1]”

Moreover, we improve our cited references and add new relevant papers that improve our background and discussion.            

“H Heitmann, TFM Andlauer, T Korn, M Muhlau, P Henningsen, B Hemmer and M Ploner [7]

M Husain and JP Roiser [8]

W Swardfager, JD Rosenblat, M Benlamri and RS McIntyre [9]

A Der-Avakian and A Markou [10]

KE Stephan, ZM Manjaly, CD Mathys, LA Weber, S Paliwal, T Gard, M Tittgemeyer, SM Fleming, H Haker, AK Seth, et al. [2]

J Savitz and NA Harrison [4]

 T McMorris, M Barwood and J Corbett [5]

A Kuppuswamy [3]

M Rottoli, S La Gioia, B Frigeni and V Barcella [6]       

A Chaudhuri and PO Behan [11] SN Haber and B Knutson [12]

Reviewer 2 Report

It needs to classify images about the sensorimotor, depression and autonomic system in brain image. The correlation between pain, emotion and autonomic imbalance was an important issue in this field.

The whole review is somewhat fragmented, and it is necessary to provide a diagram of possible pathways in the brain to give the reader an idea about the fatigue in multiple sclerosis.

Author Response

Reviewer #2

It needs to classify images about the sensorimotor, depression and autonomic system in brain image. The correlation between pain, emotion and autonomic imbalance was an important issue in this field.

ANSWER: We acknowledge that the introduction and the description of physiopathology of fatigue was not completed in the original submission. Therefore, we made some changes in the introduction and in the discussions, including the correlation of depression, pain, emotion and autonomic imbalance with fatigue. The text has been modified taking the reviewer’s comment into consideration.

“It is well-known that pro-inflammatory cytokines operate directly on the brain to induce sickness behavior, reduced motivation, increases pain sensitivity, evident fatiguability and depressed mood [1, 2]. They act affecting the monoaminergic neurotransmission and damaging the mesocorticolimbic pathways (crucial for valence and reward processing) [3].

The persistent endocrine and autonomic disturbances are likely due to gray matter lesion in hypotalamus or brainstem nuclei that could disturb the hypothalamus-pituitary-adrenal axis and descending neural control of the autonomic nervous system [4]. Indeed, the autonomic nervous system dysfunction in patients with MS, appears involved in the exacerbation of symptom of fatigue [4-8]. Dinoto et al. [9] reported a strong correlation between fatigue and autonomic nervous system dysfunction in patients with MS. Specifically they found that patients with fatigue had a significant higher dysautonomia compare to patients without fatigue. Indeed, it seems that autonomic nervous system is regulated by the same brain areas involved in the perception of fatigue. Further, the vagus nerve (the connection between interoceptive areas and autonomic involvement), is affected by pro-inflammatory cytockines and its overactivation connect symptom of fatigue and autonomic dysregulation [5]. 

Indeed, more than half of patients with MS report symptom of fatigue together with symptoms of depression and pain [3]. The coexistence of these three symptoms suggests a common etiology. Specifically, they are important sign of anhedonia (decreased ability to attempt for and to experience pleasure) [10, 11] that it has been imputed to deficits in reward processing [12] and is a central component of emotional responses, behavior and learning [3].    

An interesting one focuses only on the sensorimotor system [13]. Since patients with MS present diminished sensory attenuation, the movement execution brings the brain to conclude that the execution demands more effort than predicted [1]. This theory supposed the fatigue is a straight consequence of unexpectedly high observed effort [1].”

The whole review is somewhat fragmented, and it is necessary to provide a diagram of possible pathways in the brain to give the reader an idea about the fatigue in multiple sclerosis.

ANSWER: Thank you for the comment. We have included a schematic diagram of possible pathways of fatigue in patients with MS.

Round 2

Reviewer 2 Report

The pathways of inflammation, limbic system, mental and pain control need with a graph of brain structure and descriptions in Figure 4.

The pain pathway with sensory input and lateral inhibit control from frontal lobe, it needs detail review.

Author Response

We thank the reviewer and we have considered the comments making changes in the revised manuscript.

To facilitate the identification of the revisions, we have highlighted in red all new or modified sentences.

The pathways of inflammation, limbic system, mental and pain control need with a graph of brain structure and descriptions in Figure 4.

ANSWER: We thank the reviewer for this suggestion, we added a more specific part about the pain pathways in the Figure_4.

The pain pathway with sensory input and lateral inhibit control from frontal lobe, it needs detail review.

ANSWER: We appreciate the indication, and we thank the reviewer for the comments. Indeed, sensory input of pain and lateral inhibition from frontal lobe are an interesting point in this complicate pathophysiological phenomenon. However, the aim of the current study was to describe the structural and functional alterations in patients with MS-related fatigue. Therefore, we have attempted to address the point raised by the reviewer describing more deeply the likely correlation between the role of prefrontal cortex in the inhibition on pain and structural alterations in patients with fatigue.

“The shared etiology was demonstrated by several studies [1-6]. Seixas et al. [1], reported functional and structural alteration in the brain structures implicated in the reward circuitary in patients with MS that reported chronic pain, specifically in the caudate nucleus, the nucleus accumbens and the mesial temporal lobe. The ventral striatum, including the nucleus accumbens and the caudate nucleus, is associated with the limbic structures and the prefrontal cortex, and is implicated in motivational and emotional aspects of behaviour, including reward. Moreover, gray matter atrophy in the basal ganglia, primarily the striatum and the limbic system was shown in patients with MS that reported fatigue and depression [7].

It is important to note that, it has been demonstrated that prefrontal cortex contributes to the top-down regulation of sensory and affective processes and its projections to the periaqueductal gray, thalamus and amygdala have been demonstrated to influence chronic pain phenotypes [8, 9]. 

Zhou et al. [10], shown that inhibiting this pathways the pain getting worse, demonstrating that this pathway is employed endogenously to suppress pain. The structural alteration in prefrontal cortex reported in patients with fatigue might be related to a major sensitivity pain that enhance the symptom of fatigue.”

  1. Seixas D, Palace J, Tracey I: Chronic pain disrupts the reward circuitry in multiple sclerosis. Eur J Neurosci 2016, 44(3):1928-1934.
  2. Penner IK, Paul F: Fatigue as a symptom or comorbidity of neurological diseases. Nat Rev Neurol 2017, 13(11):662-675.
  3. Solaro C, Gamberini G, Masuccio FG: Depression in Multiple Sclerosis: Epidemiology, Aetiology, Diagnosis and Treatment. CNS Drugs 2018, 32(2):117-133.
  4. Palotai M, Guttmann CR: Brain anatomical correlates of fatigue in multiple sclerosis. Mult Scler 2020, 26(7):751-764.
  5. Schmaal L, Veltman DJ, van Erp TG, Samann PG, Frodl T, Jahanshad N, Loehrer E, Tiemeier H, Hofman A, Niessen WJ et al: Subcortical brain alterations in major depressive disorder: findings from the ENIGMA Major Depressive Disorder working group. Mol Psychiatry 2016, 21(6):806-812.
  6. Kang D, McAuley JH, Kassem MS, Gatt JM, Gustin SM: What does the grey matter decrease in the medial prefrontal cortex reflect in people with chronic pain? Eur J Pain 2019, 23(2):203-219.
  7. Heitmann H, Andlauer TFM, Korn T, Muhlau M, Henningsen P, Hemmer B, Ploner M: Fatigue, depression, and pain in multiple sclerosis: How neuroinflammation translates into dysfunctional reward processing and anhedonic symptoms. Mult Scler 2020:1352458520972279.
  8. Bushnell MC, Ceko M, Low LA: Cognitive and emotional control of pain and its disruption in chronic pain. Nat Rev Neurosci 2013, 14(7):502-511.
  9. Cardoso-Cruz H, Sousa M, Vieira JB, Lima D, Galhardo V: Prefrontal cortex and mediodorsal thalamus reduced connectivity is associated with spatial working memory impairment in rats with inflammatory pain. Pain 2013, 154(11):2397-2406.
  10. Zhou H, Martinez E, Lin HH, Yang R, Dale JA, Liu K, Huang D, Wang J: Inhibition of the Prefrontal Projection to the Nucleus Accumbens Enhances Pain Sensitivity and Affect. Front Cell Neurosci 2018, 12:240.
